# Identification and Effects of Skim Milk-Derived Bioactive Antihypertensive Peptides

**Fatah B. Ahtesh** [1], **Lily Stojanovska** [2], **Vijay Mishra** [3], **Osaana Donkor** [3], **Jack Feehan** [1], **Marijan Bosevski** [4], **Michael Mathai** [1] **and Vasso Apostolopoulos** [1,*]

1    Institute for Health and Sport, Victoria University, Melbourne 3011, Australia;
     fatah.ahtesh@live.vu.edu.au (F.B.A.); jack.feehan@vu.edu.au (J.F.); michael.mathai@vu.edu.au (M.M.)
2    Department of Nutrition and Health, College of Medicine and Health Sciences,
     United Arab Emirates University, Al Ain 999041, United Arab Emirates; lily.stojanovska@uaeu.ac.ae
3    Institute for Sustainable Industries and Liveable Cities, Victoria University, Melbourne 3011, Australia;
     vijay.mishra@vu.edu.au (V.M.); Osaana.Donkor@vu.edu.au (O.D.)
4    St. Cyril and Methodius, Faculty of Medicine, University Cardiology Clinic, 1000 Skopje, North Macedonia;
     marijanbosevski@yahoo.com
*    Correspondence: vasso.apostolopoulos@vu.edu.au

**Abstract:** Bioactive peptides are generated during milk fermentation or enzymatic hydrolysis. *Lactobacillus* (L) *helveticus* is commonly used to produce some types of fermented milk products. Fermented milk derived bioactive peptides are known to be beneficial in human health. Anti-hypertensive peptides play a dual role in the regulation of hypertension through the production of the vaso-constrictor angiotensin II and its inactivation of the vasodilator bradykinin. MALDI MS/MS, nano-LC/MS/MS and RP-HPLC were used to isolate peptides showing angiotensin converting enzyme inhibition (ACE-I) from 12% fermented skim milk using a combination of *L. helveticus* and Flavourzyme®. The fermentation procedure facilitated the identification of 133 anti-hypertensive peptides and 75% short chain amino acids, and the three with the highest ACE-I activity reduced blood pressure in a rat model of hypertension. The freeze- dried extract was supplemented in rodent chow. In this study 14-week-old male spontaneously hypertensive rats were fed for 10 weeks with the identified peptides added to chow and compared to controls supplemented with skim milk powder. Blood pressure (BP) decreased significantly ($p < 0.05$) from 6 to 10 weeks of FS groups (120/65 mmHg) compared with the NFS control groups, where the BP increased significantly (220/150 mmHg) ($p < 0.05$). The F6 fraction provided bioactive peptides with stronger antihypertensive properties than other fractions. Skim milk fermented by *L. helveticus* and Flavourzyme® generates several bioactive peptides which have a blood pressure lowering effect in hypertensive disease.

**Keywords:** anti-hypertensive peptides; Flavourzyme®; *Lactobacillus helveticus*; blood pressure; rates; fermented skim milk; Nano-LC/MS/MS; MALDI MS/MS; angiotensin-converting enzyme inhibitors





## 1. Introduction

Fermented dairy products are known to provide a large volume of nutrients and bio-functional peptides which are linked to improved health benefits [1–3]. Approximately a quarter of the world's adult population suffer from hypertension [4], of which a third are from western countries [5]. Recent studies have estimated that 1.13 billion people worldwide have hypertension, with two-thirds of those living in low- and middle-income countries. In 2015, one in four men and one in five women had hypertension [6]. Hypertension increases incidence and severity of impaired kidney function, cardiovascular disease, peripheral artery disease, heart failure and stroke [4]. Angiotensin converting enzyme inhibitory (ACE-I) peptides from the enzymatic hydrolysis of milk proteins are able to reduce the blood pressure in rat models of hypertension [7–9]. Several ACE-I peptides have

been demonstrated as products of enzymatic hydrolysis of milk proteins which are able to inhibit ACE activity in vitro [8,10–19].

In addition, reconstituted skim milk (RSM) fermented with *L. helveticus* and *Saccharomyces cerevisiae* (*S. cerevisiae*) decreases systolic blood pressure (SBP) in humans with mild hypertension ranging from 4.6–14.1 mmHg [2]. These anti-hypertensive effects have been, in part, correlated to casein-derived tripeptides released during fermentation [2]. In fact, tripeptides such as Ile-Pro-Pro (IPP) and Val-Pro-Pro (VPP) are known to improve blood pressure in humans and rats [20–22]. The ACE-I peptides have two known biological targets: the deca-peptide angiotensin-1 receptor (Asp-Arg-Val-Tyr-Ile-His-Pro-Phe-His-Leu) and 9-mer peptide bradykinin (Arg-Pro-Pro-Gly-Phe-Ser-Pro-Phe-Arg) receptor [23,24]. The two peptides Hip-Phe-Arg and hippuryl-L-histidyl-L-leucine (Hip-His-Leu) have identical binding affinity tendencies to bradykinin and angiotensin II, showing substrate specificity for ACE-I activity [23]. A number of different procedures have been utilized for the production of bioactive ACE-I peptides activity using enzymatic hydrolysis of milk proteins [25–27]. For instance, different *L. helveticus* strains are known to express proline-peptidases leading to the release of the tripeptides IPP and VPP during fermentation of milk [28,29]. However, there are no studies describing the identification of antihypertensive peptides released from fermented skim milk with *L. helveticus* combined with Flavourzyme®. As there are no data reporting the production of antihypertensive peptides from probiotic *lactobacillus* (LAB)-fermented low fat skim milk co-cultured with Flavourzyme®, the study evaluated the effect of the identified peptides on spontaneously hypertensive rats to determine their impact on high blood pressure. Results also demonstrate that the combination of LAB and Flavourzyme® significantly increases the production of antihypertensive peptides in skim milk and increases ACE-I activity. Previous published results showed that the co-culture of *L. helveticus* (ASCC 8801315) and Flavourzyme® in skim milk led to increased ACE-I activity after a relatively brief fermentation [17]. Herein, this study confirmed that fermentation of low-fat skim milk with *L. helveticus* co-cultured with Flavourzyme® significantly amplifies the release of anti-hypertensive peptides, as well as analysing the resulting peptide and amino acid profile. Rats were fed with a freeze-dried peptide for 10 weeks, after an acclimatisation period. The heart rate, blood pressure and food intake of the rats were measured daily, and body weight measured weekly.

## 2. Results and Discussion

Fermentation with *L. helveticus* (881315) and Flavourzyme® causes significant proteolysis.

The micro-fluidic Loa-C method provides a unique approach to the real-time separation of major milk proteins as well as giving information on size, concentration and purity of proteins in a single assay [30]. The simulated gel patterns for fermented and un-treated skim milk as control, obtained by Loa-C with the elution profiles are shown (Figure 1). Preliminary studies making use of the Agilent Protein 240 kit indicated its suitability for the separation of most milk proteins (Figure 1A–C); migration distribution of proteins in control RSM and fermented RSM are shown in Figure 1B,C. Un-hydrolysed RSM showed groups of proteins together as 2 peaks at 28 kDa and 46 kDa (Figure 1B), indicating the un-hydrolysed milk proteins [31]. The migration pattern and profile shows a number of peaks ranging from 5–240 kDa in fermented milk. This indicates the products of the casein and other milk protein hydrolysates with a range of molecular weights of the resulting peptides.

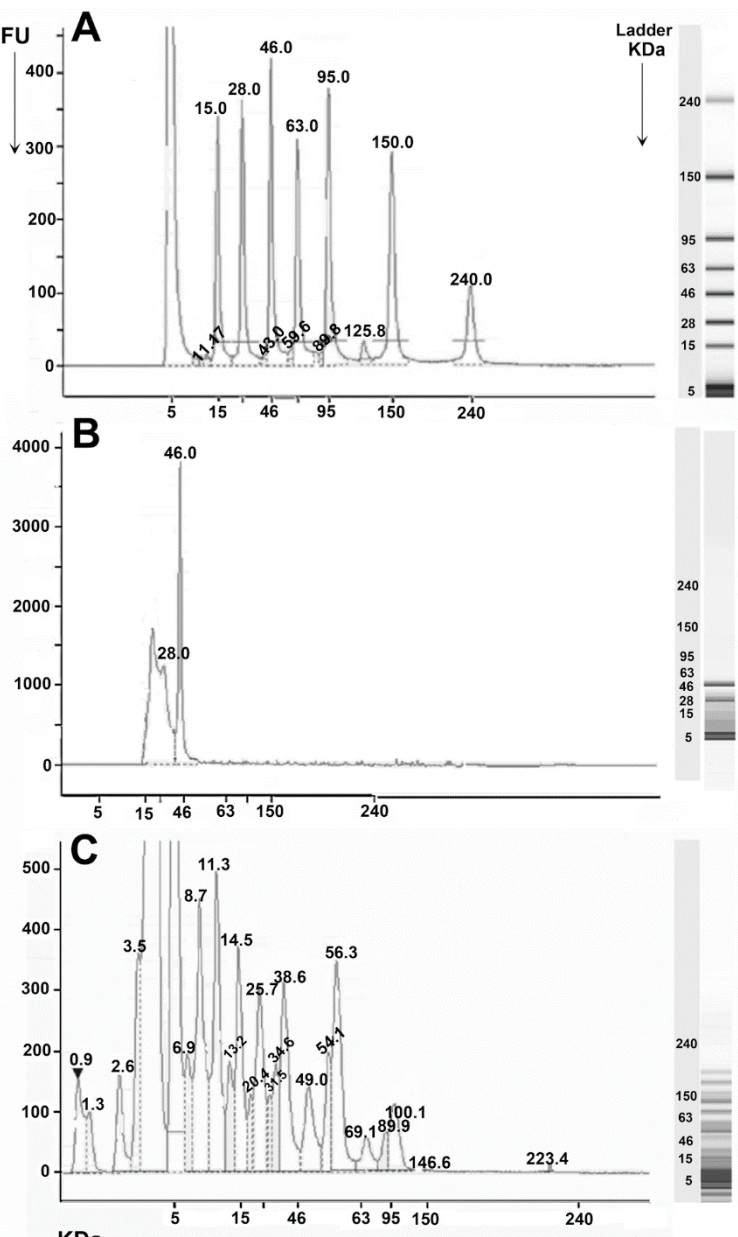

**Figure 1.** Lab-on-a-chip capillary electrophoresis with elution profiles. Migration pattern for (**A**) proteins, (**B**) non-treated (control) skim milk proteins and (**C**) fermented skim milk protein hydrolysis from the combination of *L. helveticus* 881315 and Flavourzyme® following incubation for 12 h at 37 °C. Molecular weights (ladder; KDa) are shown.

Fermentation with the combination of *L. helveticus* and Flavourzyme® causes increased hydrolysis versus either independently.

The degree of hydrolysis (DH) of RSM following combined hydrolysis with *L. helveticus* (881315) and Flavourzyme® were compared to hydrolysis of *L. helveticus* (881315) or Flavourzyme® alone after 12 h of fermentation at 37 °C (Figure 2). The hydrolysis of the combined cultures increased rapidly in the first 1 h of fermentation and was significantly increased after 8 h compared to Flavourzyme® or *L. helveticus* (881315) alone ($p < 0.05$). No apparent hydrolysis was observed after this period. However, the DH after 12 h fermentation was highest for the combination of *L. helveticus* (881315) and Flavourzyme® with 70.9% hydrolysis (Figure 2). The DH reached ~20% and ~10% during the 12 h fermentation for *L. helveticus* 881315 or Flavourzyme® alone, respectively. This is similar to what others have reported during 6 h fermentation using proteases from *B. licheniformis* [32]. The DH

significantly increased with the combination of *L. helveticus* (881315) and Flavourzyme[®] as well as with increased fermentation time (*p* < 0.05) (Figure 2). The data suggest that Flavourzyme[®]-supplementation has the greatest effect on the DH of skim milk protein. The significant increase in the total DH by the combined activity was likely due to complementary substrate specificities, resulting in improved proteolytic activity [17]. These results were expected considering our previously published data, which showed increased bioactivities of *L. helveticus* in combination with Flavourzyme[®] during 12 h fermentation compared to the strain alone [17,33]. High proteolytic activity correlates with high DH and this was reflected in the performance of the combined effect of *L. helveticus* (881315) and Flavourzyme[®], causing a significant increase in bioactivities of ACE-I peptide production [34].

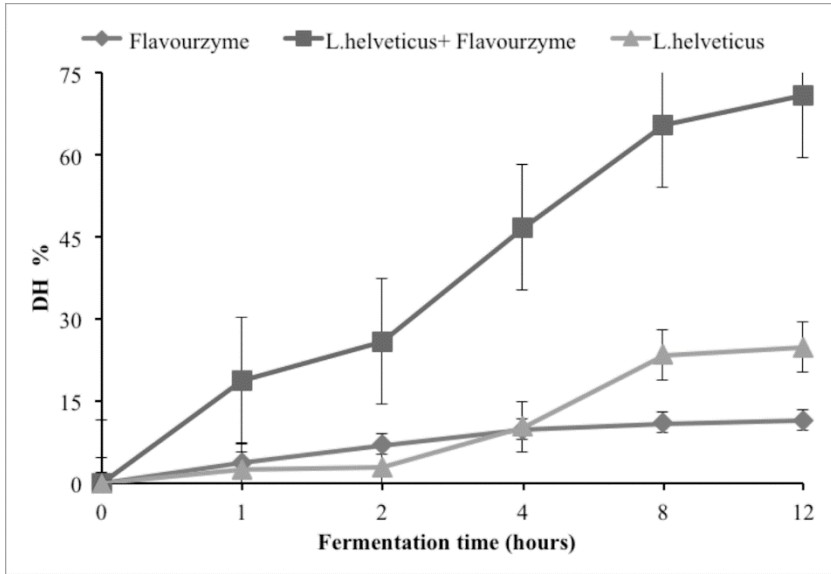

**Figure 2.** Degree of hydrolysis (DH%) in fermented skim milk generated following the combination of *L. helveticus* 881315 and Flavourzyme[®] (square), *L. helveticus* 881315 alone (triangle), and Flavourzyme[®] alone (diamond) at 37 °C for 0–12 h fermentation. Values are mean ± SEM of three determinations for DH values.

Identification of specific bioactive peptides after fermentation with *L. helveticus* (881315) and Flavourzyme[®].

Of the 6 fractions, F1 and F6 showed the highest ACE-I bioactivity between 85.40% and 95.51%, respectively, with the lowest $IC_{50}$ of 0.01 mg/mL (Table 1 and Figure 3A). The ACE-I bioactivities of F1 and F6 were significantly higher (*p* < 0.05) than the others, with both having an $IC_{50}$ of 0.01 mg/mL, followed by F2 (72.04% and $IC_{50}$ 0.34 mg/mL). The activity of the remaining fractions was considered low or minimal. LC/MS analysis of fractions F1 and F6 showed several peptide components (Tables 2 and 3, Figure 3A) and peptide fraction profiles (Figure 3B,C). The MS results showed that sample F6 contained 109 unique peptides which matched known peptides with 99% confidence (*p* < 0.05) from the bovine database (Table 3) and showed the highest ACE-I peptide activity (Table 1). In F1, 24 unique peptides were identified with greater than 99% confidence. When the bacterial database (Swiss Prot 2013) was searched, the mass error 55.7 ppm was higher than all peptides matched compared to the bovine database. Peptides greater than 1150 Da (214–224) were isolated from F6 with casein origin (Tables 2 and 3). The LCMS/MS spectrum matched one sequence of the group based on mass from milk casein (Figure 3). The major fragment ions were noted between *m/z* 903.30–2898.01 and 1702.96–2108.21 for F1 and F6, respectively, and were classified as β-type ions next to proline, primarily β3 and β5, respectively. Proline is commonly associated with large volumes of y- and β-type fragment ions due to cleavage of peptide bonds adjacent to the amino acid. The bulk of the peptide components of each HPLC fraction were identified and the results are summarized in

(Tables 1 and 3). Peptides containing amino acids such as proline, aspartic acid, tryptophan, cysteine, methionine, tyrosine, leucine, arginine, histidine and alanine have been noted to confer higher anti-hypertensive activities [3,35]. As seen in Figure 3, for example, a purified peptide identified in F6 TPVVVPPF was located between *f* 10–42. Most of the peptides in F6 contained the proline amino acid, which has been shown to have effects similar to captopril in activity [15,36] and is high in ACE-I bioactivities [3] and has anti-hypertensive properties. The anti-hypertensive peptide FFVAPFPGVFGK was identified in F6 (Table 3). This peptide has also been shown to reduce blood pressure in spontaneously hypertensive rats [37]. In addition, peptides with anti-oxidative and anti-bacterial properties were also present in the peptide sequences, such as YQEPVLGPVRGPFPIIV and SLPQNIPPLTQTPVVVPPF (Table 3) [38,39] and also anticancer peptides such as NLHLPLPLL, ENLHLPLPLL and VENLHLPLPLL and antioxidant peptide such as VAMVPPFET and GQVPP (Table 3), as matched and identified in the database. Amongst the identified peptides, GPVRGPFPIIV, LHLPLPLL, YQKFPQY, AYFYPEL and RYLGY had significant $IC_{50}$ values (0.7–6.5 mM). These peptides have been previously shown to have anti-hypertensive properties in spontaneous hypertensive rats, with RYLGY and AYFYPEL having similar activity to that of the tripeptide VPP when orally administered [40]. Table 3 shows LVYPFPGPIPNSLPQNIPP, LVYPFPGPIPNSLPQN, TPVVVPPFLQP and LTQTPVVVPPF as ACE-Inhibitors from the database as indicators. In both fractions a high amount of β-lacto globulin was also present. β-lacto globulin has high biological importance as a source of bioactive peptides [41–43]. Overall, in this study, F6 contained more than 75% short chain amino acid peptides, with the highest ACE-I bioactivities compared to other fractions. The other 25% of peptides have previously been identified as antioxidative, anti-cancer and as conferring improved immunity [22,35,44].

Peptides produced by Fermentation with *L. helveticus* (881315) and Flavourzyme® have a strong anti-hypertensive effect in spontaneously hypertensive rats.

**Table 1.** The percentage ACE-inhibitory activity and $IC_{50}$ mg/mL (means ± SE) of fermented 12% skim milk peptide fractions.

| Fractions | ACE-I % | $IC_{50}$ mg/mL |
|:---:|:---:|:---:|
| F1 | 85.40 ± 0.32 [a] | 0.01 ± 0.00 [d] |
| F2 | 72.04 ± 0.91 [b] | 0.34 ± 0.03 [cd] |
| F3 | 42.39 ± 1.20 [c] | 0.47 ± 0.09 [bc] |
| F4 | 17.31 ± 2.21 [d] | 1.18 ± 0.16 [a] |
| F5 | 27.91 ± 5.30 [d] | 0.78 ± 0.11 [b] |
| F6 | 95.51 ± 0.21 [a] | 0.01 ± 0.00 [d] |

Values followed by different letters indicate significant differences between fractions *p* < 0.05.

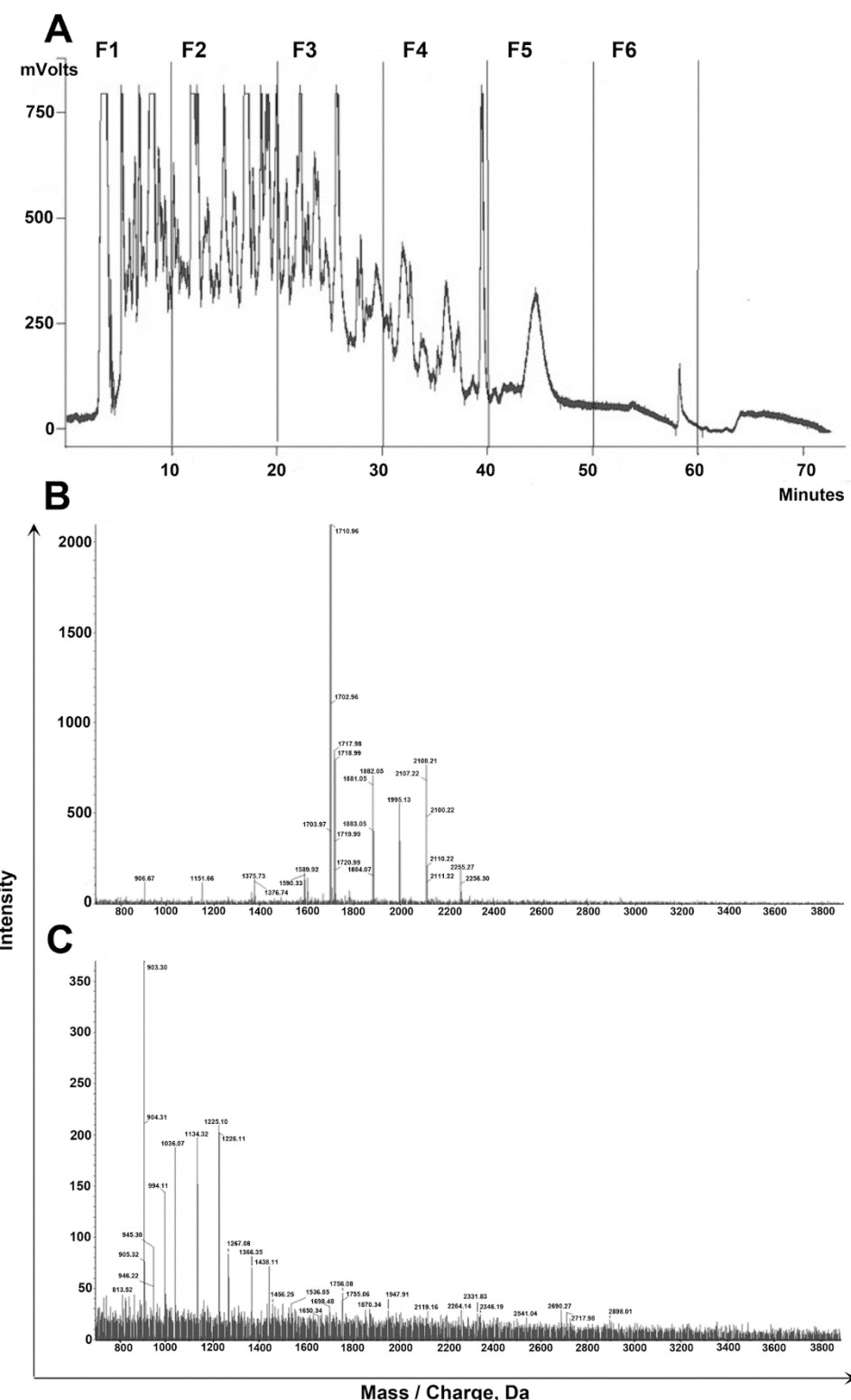

**Figure 3.** (**A**) Identification of 6 peptide fractions (F1-F6) by RP-HPLC from skim milk fermented with a combination of *L. helveticus* 881315 and Flavourzyme®. Molecular mass for (**B**) F1 and (**C**) F6 are shown.

**Table 2.** Protein families of peptides identified in fraction 1 (F1) and 6 (F6).

| Protein Family | Mass (Da) | Sequence |
|---|---|---|
| Fraction 1 (F1) | | |
| CASB_BOVIN | 25,091 | Beta-casein |
| CASA1_BOVIN | 24,513 | Alpha-S1-casein |
| CASA1_BUBBU | 24,311 | Alpha-S1-casein |
| LACB_BOVIN | 19,870 | Beta-lacto globulin |
| CAC1E_RABIT | 254,089 | Voltage-dependent R-type calcium channel subunit alpha |
| CASA2_CAPHI | 26,372 | Alpha-S2-casein |
| LALBA_BOSMU | 16,237 | Alpha-lactoalbumin |
| Fraction 6 (F6) | | |
| CASB_BOVIN | 25,091 | Beta-casein |
| CASA1_BOVIN | 24,513 | Alpha-S1-casein |
| CASA1_BUBBU | 24,311 | Alpha-S1-casein |
| LACB_BOVIN | 19,870 | Beta-lacto globulin |
| CASA2_BOVIN | 26,002 | Alpha-S2-casein |
| LALBA_BOSMU | 16,237 | Alpha-lactoalbumin |
| CASK_BOVIN | 21,256 | Kappa-casein OS |
| FETUA_BOVIN | 38,394 | Alpha-2-HS-glycoprotein |
| GLCM1_BOVIN | 17,141 | Glycosylation-dependent cell adhesion molecule |
| DDX56_BOVIN | 61,216 | Probable ATP-dependent RNA helicase |
| BRAT1_AILME | 88,137 | BRCA1-associated ATM activator |
| NIF3L_BOVIN | 41,880 | NIF3-like protein |

Systolic blood pressure (SBP) and diastolic blood pressure (DBP) after 10 weeks of dietary supplementation are shown in Figure 4. The BP values of all groups were similar at baseline (W0; ~155/80 mmHg). There was a decrease in SBP and DBP after oral administration of the peptide extract of *L. helveticus* (881315) and Flavourzyme®-fermented skim milk (FS) at week 10 (W10), compared with controls. At the end of the feeding period the rats fed FS showed a 40% (120 mmHg) decrease in SBP, and a 30% (65 mmHg) decrease in DBP, which is typically considered to be a normal blood pressure level for a rat (Figure 4). The group fed NFS showed an increase in both SBP and DBP (220 mmHg and 150 mmHg respectively) as is described in the SHR model. It is likely that the milk protein-derived peptides with ACE inhibiting properties in the FS fed group caused the reduction in BP [45]. It was found that the ACE-I activity was higher (95.5%) in peptides containing short chain amino acids, such as TPVVPPF, YPFPGPIP and SLPQNIPPLTQTPVVPP (Table. 3). Consistent with this, in a study of hypertensive patients, sour milk fermented by *L. helveticus* and *Saccharomyces cerevisiae* reduced systolic and diastolic blood pressure over an 8-wk intervention. A milk product fermented by *L. helveticus* LBK-16H has been shown to have an effect on blood pressure both in animal models and in humans [46]. In an 8-wk study, the fermented milk product containing the bioactive tripeptides Val-Pro-Pro and Ile-Pro-Pro reduced blood pressure in mildly hypertensive subjects [46]. Similar peptides, including tripeptides, have been reported to have anti-hypertensive properties to that of the common medication Captopril [8]. Additionally, Nakamura, Yamamoto, Sakai, Okubo, Yamazaki and Takano [24] suggested that peptides in fermented milk have both immediate and long-lasting anti-hypertensive effects in rats. Small di- or tripeptides are easily absorbable in the digestive tract [44,47], and the Pro-Pro sequence resists degradation by digestive enzymes [48–50] making these peptides likely to enter the circulation whole. A likely hypothesis for the in vivo action of ACE-I peptides is that directly absorbed active tripeptides reach the systemic circulation where they encounter and competitively inhibit ACE. This is supported by studies showing that administration of low molecular weight peptides attenuates and even reverses the progression of hypertension in rats [9].

Another study suggested that about 90% of the absorption of peptides in the gastrointestinal tract takes place in the region of the small intestine [51]. The in vitro evidence of the properties of the ACE-I peptides are supported by a reduction in the activity of ACE alongside decreased circulating angiotensin-II and aldosterone and an increase in renin as a compensatory feedback mechanism in rats, all of which suggests ACE as the target of these peptides in vivo [9]. Plasma renin activity and level has also been shown to be elevated in hypertensive rats after 14 weeks of supplementation with IPP and VPP [28]. This increase is suggested to be due to a lack of negative regulation by angiotensin II, supporting the theory of ACE inhibition [52–54]. It has also been shown that ingestion of milk fermented with *L. helveticus* partially attenuates the blood pressure response to angiotensin-I in unconscious rats, with a concomitant increased response to bradykinin, further supporting the inactivation of ACE as the underlying mechanism [40]. While promising, the mechanistic theory of ACE inhibition by tripeptides is yet to be validated, and a broad range of confounding effects have to be considered [29]. These results show the effects and robustness of biologically active antihypertensive peptide production during fermentation with the combination of *L. helveticus* and Flavourzyme®.

**Table 3.** Identification of peptides produced from fermented 12% RSM in combination of *L. helveticus* 8801315 and Flavourzyme® isolated from RP-HPLC in fractions 6 (F6) and F1 using MALDI-MS/MS analysis. The mascot database search program used was used.

| Protein Accession | Protein Description | Peptide, *m/z* (Experimental) | Peptide Mass, Da (Experimental) | Peptide Mass, Da (Calculated) | Peptide Sequence |
|---|---|---|---|---|---|
| Fraction 6 (F6) CASB_BOVIN | Beta-casein | 458.3093 | 914.6041 | 914.5953 | LHLPLPLL |
| CASB_BOVIN | Beta-casein | 522.3336 | 1042.6526 | 1042.6539 | LHLPLPLLQ |
| CASB_BOVIN | Beta-casein | 576.3528 | 1150.691 | 1150.6863 | GPVRGPFPIIV |
| CASA1_BOVIN | Alpha-S1-casein | 552.8222 | 1103.6299 | 1103.6339 | LGYLEQLLR |
| CASA1_BOVIN | Alpha-S1-casein | 559.3271 | 1116.6396 | 1116.6291 | VLNENLLRF |
| CASA1_BOVIN | Alpha-S1-casein | 571.7766 | 1141.5387 | 1141.5251 | SDIPNPIGSEN |
| CASA1_BOVIN | Alpha-S1-casein | 609.3681 | 1216.7216 | 1216.7179 | LGYLEQLLRL |
| CASA1_BOVIN | Alpha-S1-casein | 610.3212 | 1218.6279 | 1218.6285 | VAPFPEVFGKE |
| CASA1_BOVIN | Alpha-S1-casein | 669.8782 | 1337.7419 | 1337.7191 | GLPQEVLNENLL |
| CASA1_BOVIN | Alpha-S1-casein | 683.8746 | 1365.7347 | 1365.6969 | FFVAPFPEVFGKE |
| CASA1_BUBBU | Alpha-S1-casein | 559.3271 | 1116.6396 | 1116.6291 | VLNENLLRF |
| CASA1_BUBBU | Alpha-S1-casein | 571.7766 | 1141.5387 | 1141.5251 | SDIPNPIGSEN |
| CASA1_BUBBU | Alpha-S1-casein | 609.3681 | 1216.7216 | 1216.7179 | LGYLEQLLRL |
| CASA1_BUBBU | Alpha-S1-casein | 610.3212 | 1218.6279 | 1218.6285 | VAPFPEVFGKE |
| CASA1_BUBBU | Alpha-S1-casein | 683.8746 | 1365.7347 | 1365.6969 | FVAPFPEVFGKE |
| Fraction 1 (F1) CASA1_BOVIN | Alpha-S1-casein | 24,513 | 1828.8711 | 1828.8513 | DIPNPIGSENSEKTTMP |
| CASA1_BOVIN | Alpha-S1-casein | 24,513 | 997.5195 | 997.508 | GLPQEVLNE |
| CASB_BOVIN | Beta-casein | 25,091 | 1258.6949 | 1258.6921 | DVENLHLPLPL |
| CASB_BOVIN | Beta-casein | 25,091 | 1332.6589 | 1332.6424 | EMPFPKYPVEP |
| CASB_BOVIN | Beta-casein | 25,091 | 1150.691 | 1150.6863 | GPVRGPFPIIV |
| CASB_BOVIN | Beta-casein | 25,091 | 1356.8285 | 1356.8017 | IPPLTQTPVVVPP |
| CASB_BOVIN | Beta-casein | 25,091 | 1503.8765 | 1503.8701 | IPPLTQTPVVVPPF |
| CASB_BOVIN | Beta-casein | 25,091 | 1503.8817 | 1503.8701 | IPPLTQTPVVVPPF |
| CASB_BOVIN | Beta-casein | 25,091 | 1363.7167 | 1363.6846 | KEMPFPKYPVE |
| CASB_BOVIN | Beta-casein | 25,091 | 1380.7026 | 1380.6972 | MHQPHQPLPPTV |
| CASB_BOVIN | Beta-casein | 25,091 | 1219.619 | 1219.5947 | MPFPKYPVEP |
| CASB_BOVIN | Beta-casein | 25,091 | 1319.7531 | 1307.7085 | PVVVPPFLQPEV |

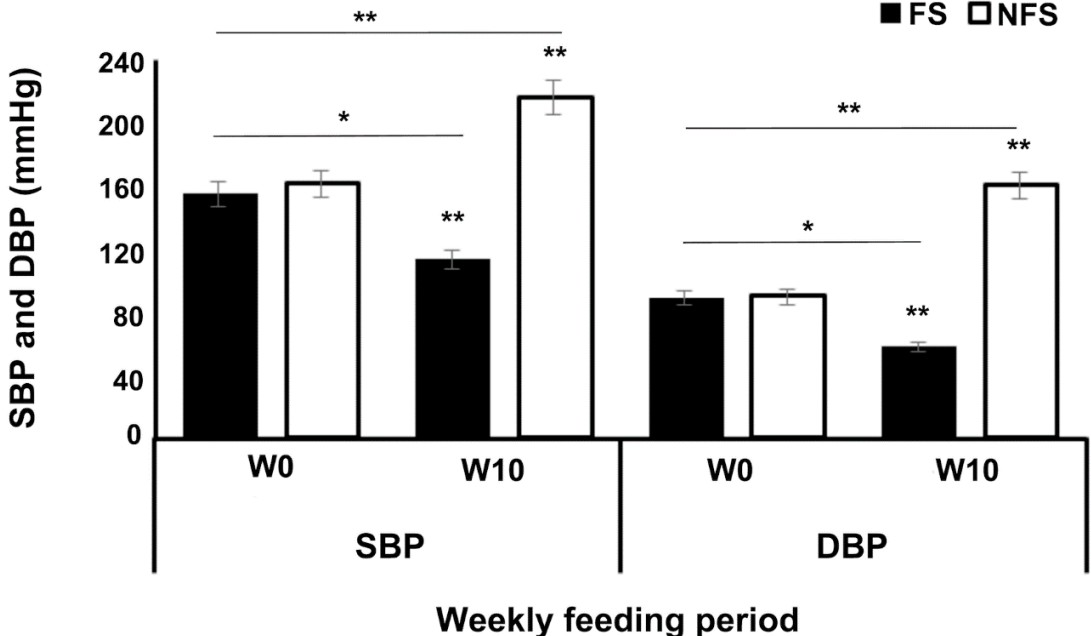

**Figure 4.** Systolic blood pressure (SBP) and diastolic blood pressure (DBP) of spontaneously hypertensive rats (SHR). Oral administration (10 weeks) of fermented skim milk containing peptides (FS) compared with skim milk powder as control (NFS). Error bars represent standard error of the means (±SEM). * $p < 0.05$, ** $p < 0.01$.

## 3. Conclusions

This study successfully identified 133 bioactive peptide candidates with 99% confidence from 2 fractions (F1 and F6) resulting from the fermentation of reconstituted skim milk with *L. helveticus* and Flavourzyme®. The peptides with the highest ACE-I bioactivity were in F6 (95.51% with $IC_{50}$ 0.01 mg/mL). Dietary supplementation with the peptides extracted from fermented skim milk exhibited potent antihypertensive peptides leading to the normalisation of SBP and DBP in spontaneously hypertensive rats during 10 weeks of feeding, compared to the controls. This information could pave the way to the development of fermented, antihypertensive milk products to be used to assist with the prevention and management of the growing burden of hypertension.

## 4. Materials and Methods

### 4.1. Chemicals

The following chemical reagents were used in experiments: Trichloroacetic acid (TCA), sorbitol, trifluoroacetic acid (TFA), β-mercaptoethanol, tris-hydrochloric acid, glycerol, dithiothreitol and acetic acid (Sigma-Aldrich, Melbourne, Australia), acetonitrile (Merck, Darmstadt, Germany), bacteriological peptone (Oxoid, Melbourne, Australia). O-phthaldialdehyde, tripeptides hippuryl-histidyl-leucine, bacteria media de Man Rogosa and Sharpe (MRS) and agar (Sigma-Aldrich; Melbourne, Australia). Skim milk powder containing 1.2% fat, 37% protein, 8.6% ash and 52% lactose (Devondale Murray Goulburn, Melbourne, Australia); protease enzymes from Aspergillus (Flavourzyme®; EC 3.4.11.1) (Novozymes, North Rocks, Australia), *L. helveticus* (ASCC 881315) (Dairy Innovation, Werribee, Australia).

### 4.2. Propagation of Cultures and Preparation of Fermented RSM

Activation of cultures was performed according to previously published methods with minor modifications [17,18]; MRS broth (10 mL) was mixed with 1% culture for 18 h at 37 °C. *L. helveticus* (881315) (1% *v/v*) and 0.14% constituted Flavourzyme® were added into 10 mL RSM (12% *w/v* in distilled water). Following two transfers, cultures were added into

fresh RSM. The RSM was pasteurised via heat treatment for 30 min at 90 °C and cooled to 42 °C, after which the fermentation process started using a bioreactor fermentation system.

### 4.3. Bioreactor Assay of RSM

A 5L bioreactor (Bio-Stat® A plus, Goettingen, Germany) was used for fermentation of 12% pasteurized RSM combined with *L. helveticus* (881315) and Flavourzyme® at 37 °C for 12 h. The bioreactor (jacketed thermostatic water bath) was maintained at a constant temperature. Milk was stirred continuously by impellers at 250 rpm and the pH was monitored with a sterile electrode (DPAS Ingold, Paris, France) and a digital transmitter (Demca 3B 1015; Alfortville, France). These were calibrated prior to media inoculation. Bacterial growth rate and pH were monitored, and ACE-I and proteolytic activities were determined for up to 12 h of fermentation.

### 4.4. Determination of Hydrolysis and ACE-I Activity

The degree of hydrolysis is defined as the number of peptide bonds broken ($\alpha$-amino nitrogen) divided by the total number of bonds (number of protein nitrogens) $\times$ 100% in the substrate as described by [55]. The method for the assessment of the degree of hydrolysis was previously published [17,18,33]. The determination of half maximal inhibitory concentration ($IC_{50}$) and percentage ACE-I activity of the produced peptides was carried out as previously described [17,18,33].

### 4.5. Micro-Fluidic Lab-on-a-Chip Electrophoresis

The bio-analyser (Agilent 2100; Agilent Technologies, Wald Bonn, Germany), was used to perform the micro-fluidic lab-on-a-chip (Loa-C) using a high sensitivity protein 250 kit and 2100 software. Samples, dye and the preparation of chip were carried out as previously described [30] with minor alterations. Briefly, a reconstituted dye solution (0.5 μL) was mixed with a 5 μL protein molecular weight ladder (5–240 kD) and 5 μL of sample in micro tubes and mixed before being incubated on ice for 30 min. The tubes were heated at 95 °C for 5 min and cooled for 15 s. Distilled water was added to the samples to generate a 90 μL final volume. All samples were thoroughly mixed and incubated on ice before analysis via a new primed chip.

### 4.6. Isolation and Characterization of Peptides

Isolation of ACE-I peptides by RP-HPLC. RSM (12% *w/v*) was fermented with *L. helveticus* (881315) and Flavourzyme® as described above, followed by centrifugation for 30 min at 14,000× *g* (Sorvall RT7, Newtown, CT, USA) and freeze drying of the supernatant. The resulting material (80 mg) was dissolved in 1 mL of solvent (0.1% TFA in deionised water), passed through a membrane filter (0.2 μm) and analysed by reverse-phase high performance liquid chromatography (RP-HPLC). Sample (1 mL) was injected onto a RP-column C-18 Jupiter Proteo 90A 250 mm × 10.0 mm, 10 micron (Phenomenex, Sydney, NSW, Australia). The mobile phase constituted solvent A (0.05%) and solvent B (60%) made up of 0.1% TFA in 90% *v/v* acetonitrile in distilled water. A linear gradient of 0–100% solvent A and >90% solvent B were used to elute the samples at a steady 1 mL/min flow rate. Elution profiles of the samples were determined by a UV detector (214 nm) for 70 min. The RP-HPLC separation procedure of samples was repeated 15 times and fractions pooled together into 6 tubes to obtain higher concentrations (F1–F6). A vacuum evaporator was used to concentrate the fractions and ACE-I activity determined as explained in our previous work [18]; those with the highest activity were further purified and analysed.

### 4.7. ACE-I Peptide Identification

Nano-LC/MS/MS analysis. Freeze dried peptide fractions which yielded the highest ACE-I (F1 and F6) activity underwent nano-LC/MS/MS analysis at the Australian Proteome Analysis Facility, NSW, Australia. In order to characterize the peptide sequences, they were searched against the Swiss-Prot protein database (539,829 sequences, 2013).

Other bacterial and mammalian species were searched (13,034 and 328,828 sequences in the Swiss-Prot database, respectively) and chosen for taxonomic categorization. Precursor and product ion mass tolerances were set at 300 ppm and $\pm$ 0.6 Da, respectively. Enzyme restriction was set as none with an allowance of one missed cleavage. Methionine oxidation was set as a variable modification.

### 4.8. Matrix Assisted Laser Desorption Ionization (MALDI)-MS/MS Analysis

Matrix was arranged by dissolving alpha-cyano-4-hydroxycinnamic acid (1 mg/mL in 90% acetonitrile, 9.9% water 0.1% formic acid). Samples were zip-tip extracted and spotted onto a MALDI target plate. For calibration of the MS, samples of bradykinin, angiotensin-I and adrenocorticotropic hormone sequence 18–39 were used. MALDI-MS/MS was performed using the 4800 plus MALDI TOF/TOF Analyser (AB Sciex, Framingham, MA, USA). For identification of proteins from the resultant data, text files were generated from 2d files and searched using the Swiss-Prot protein database (539,829 sequences, 2013). Other bacterial and mammalian species were searched (13,034 and 328,828 sequences in the Swiss-Prot database, respectively) and chosen for taxonomic categorization. Search results were generated with a significance threshold of ($p < 0.01$) with a minimum cut-off of 24 for all samples except for F1 ($p < 0.05$) with a cut-off of 60.

### 4.9. Animal Experiments

This study was performed with approval of Victoria University's Animal Ethics Committee (AEC 12/009). Spontaneously hypertensive rats (SHR), male ($n = 18$; aged 14 weeks age) were purchased from the Animal Resource Centre, Western Australia. The animals were housed at the Western Centre for Health Research and Education, Western Health, Melbourne, Australia. The weight of the rats was recorded upon arrival (weight, $250 \pm 5$ g). Three SHR were housed per stainless steel wire mesh cage in a 12:12 h light:dark cycle. The rats were divided into 2 groups ($n = 9$) and were allowed free access to feed and water. All animals were fed with certified standard laboratory rat chow (Specialty feeds, Glen Forrest, WA, Australia). The experimental group of rats (FS) were fed diets supplemented with freeze dried fermented skim milk containing the isolated anti-hypertensive peptides, with the control group (NFS) fed untreated pelleted skim milk with chow. During the 10-week experimental period, the animals were fed daily with new supplemented investigational food pellets, with access to both food and tap water ad libitum.

Blood pressure of the hypertensive rats was measured weekly by tail-cuff sphygmomanometer by volume pressure recording (VPR) sensor technology (CODA$^\circledR$ non-invasive blood pressure system NIBP, Kent scientific corporation, Inc., Torrington, CT, USA). Rats were adapted for three weeks and trained on a hypertensive monitor system, CODA NIBP. Over three weeks, rats were fed on normal standard chow (40–70 g/box of pellets daily). Daily feed intake, hypertension and heart rate of all rats were measured, and the body weight of rats were recorded weekly.

### 4.10. Statistical Analyses

Using Minitab 16 (Minitab LLC, State College, PA, USA), data were calculated as mean of 3 replicates $\pm$ standard error of the mean (SEM). Variances within the experimental groups were identified by one-way ANOVA with mean differences tested by Tukey's test. Significance level was set as $p \leq 0.05$.

**Author Contributions:** Conceptualization, L.S., V.A. and M.M.; methodology, F.B.A., V.M. and O.D.; formal analysis, F.B.A. and M.B.; investigation, F.B.A.; resources, V.A. and M.M.; writing—original draft preparation, F.B.A. and J.F.; writing—review and editing, L.S., V.M., O.D., M.B., M.M. and V.A.; visualization, F.B.A.; supervision, V.A., L.S. and M.M. All authors have read and agreed to the published version of the manuscript.

**Funding:** This research received no external funding.

**Institutional Review Board Statement:** The study was conducted according to the guidelines of Victoria University's Animal Ethics Committee and approval code was AEC 12/009.

**Data Availability Statement:** Data available on request.

**Acknowledgments:** F.B.A. was supported by scholarship funds from the Government of Libya and from the College of Health and Biomedicine, Victoria University, VIC Australia. V.A. was supported by the College of Health and Biomedicine start-up funds and the Centre for Chronic Disease. All authors were also supported by the Institute for Health and Sport, Immunology and Translational Research Group, Victoria University, Australia.

**Conflicts of Interest:** The authors declare no conflict of interest.

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
