# Peer review of "Identification and Effects of Skim Milk-Derived Bioactive Antihypertensive Peptides"

_biologics, doi:10.3390/biologics2010001_

Round 1

Reviewer 1 Report

I am convinced that the manuscript by Fatah B Ahtesh  and co-workers provides lots of valuable data and after some explanations and improvements could be considered in Biologics. But after explanations regarding issues raised below.

Some remarks:

Abstract:

Lines 14 and 15: Anti-hypertensive peptides – avoid repetition

Please provide rationale for a combinatory use of L.helveticus and Flavourzyme®. You stated that the former is commonly used.

Please provide a clear explanation why did you used a skim-milk.

Line 23-25: please explain the groups in the experiment.

The F6 fraction – explain

BP, explain

The last sentence in the abstract section should be less assertive?

Line 25: you cannot say that in the NFS the BP increased significantly (220 /150 mmHg) (P < 0.05). You had not the control group without milk addition. You are allowed to compare two groups only.

The experimental design:

Main concerns:

The experimental dietary treatments are not properly constructed: in my opinion the use of commercial diet – “the normal standard chow” (aimed as feed for rodents before or after a nutritional experiment) is to some extent wrong. The first: the different batches of that feed produced may differ substantially. The second: the authors should prepare semi-purified laboratory diets containing well-analyzed ingredients, casein (or soya as the sole protein source), fat, starch, vitamin mixture, mineral mixture, etc. In my opinion in such way the experiment could be repeated in an another lab. Please, explain.

Statistics: when you compare two groups only, you cannot use one-way ANOVA but for instance t-test. Verify, explain.

Author Response

We thank the reviewer for their feedback - please see the attached for a point-by-point response. 

Reviewer 2 Report

Dear Authors,

The manuscript is written at a very high professional level, therefore, I have found some minor editorial rather than methodical errors. The ms is adequately divided into individual chapters, the abstract is concise, the methodology is described accordingly, the data are properly statistically processed, clearly presented, and adequately commented in the discussion. The conclusions of the study were described correctly and are based on the results obtained earlier.

The manuscript is well written and its contents are appropriate but I noticed some editorial errors:

The use of latin names: If you used latin name in the introduction e.g. Lactobacillus helveticus, then you should use the short version  L. helveticus. I think it is stated in the guide for authors of the Journal. See line 196

Table 1 –some letters are not in superscript (in IC50, F2 and F3)

Some flaws in the literature:

Nb 6 – there is no access data

Nb 23 – unnecessary captitalization of the surnames

In all references cited there is lack of the name of the journal. Please correct. And some dots are missing, initials are not capitalized etc.

At the end I would like to congratulate the Authors of excellent study.

Author Response

We thank the reviewer for their feedback. Please see the attached document for a point-by-point response

Round 2

Reviewer 1 Report

I accept the revision and recommend the paper to be published.